# Transcriptomic Analyses of Grapevine Leafroll-Associated Virus 3 Infection in Leaves and Berries of ‘Cabernet Franc’

**DOI:** 10.3390/v14081831

**Published:** 2022-08-21

**Authors:** Yashu Song, Robert H. Hanner, Baozhong Meng

**Affiliations:** 1Department of Molecular and Cellular Biology, University of Guelph, Guelph, ON N1G 2W1, Canada; 2Department of Integrative Biology and Biodiversity Institute of Ontario, University of Guelph, Guelph, ON N1G 2W1, Canada

**Keywords:** grapevine leafroll-associated viruses, *Ampelovirus*, *Closteroviridae*, *Vitis vinifera*, RNA-Seq, transcriptomic analysis, RT-qPCR, photosynthesis, sugar transport

## Abstract

Grapevine leafroll-associated virus 3 (GLRaV-3) is one of the most important viruses affecting global grape and wine production. GLRaV-3 is the chief agent associated with grapevine leafroll disease (GLRD), the most prevalent and economically destructive grapevine viral disease complex. Response of grapevine to GLRaV-3 infection at the gene expression level is poorly characterized, limiting the understanding of GLRaV-3 pathogenesis and viral-associated symptom development. In this research, we used RNA-Seq to profile the changes in global gene expression of Cabernet franc, a premium red wine grape, analyzing leaf and berry tissues at three key different developmental stages. We have identified 1457 differentially expressed genes (DEGs) in leaves and 1181 DEGs in berries. The expression profiles of a subset of DEGs were validated through RT-qPCR, including those involved in photosynthesis (VvPSBP1), carbohydrate partitioning (VvSUT2, VvHT5, VvGBSS1, and VvSUS), flavonoid biosynthesis (VvUFGT, VvLAR1, and VvFLS), defense response (VvPR-10.3, and VvPR-10.7), and mitochondrial activities (ETFB, TIM13, and NDUFA1). GLRaV-3 infection altered source–sink relationship between leaves and berries. Photosynthesis and photosynthate assimilation were inhibited in mature leaves while increased in young berries. The expression of genes involved in anthocyanin biosynthesis increased in GLRaV-3-infected leaves, correlating with interveinal tissue reddening, a hallmark of GLRD symptoms. Notably, we identified changes in gene expression that suggest a compromised sugar export and increased sugar retrieval in GLRaV-3-infected leaves. Genes associated with mitochondria were down-regulated in both leaves and berries of Cabernet franc infected with GLRaV-3. Results of the present study suggest that GLRaV-3 infection may disrupt mitochondrial function in grapevine leaves, leading to repressed sugar export and accumulation of sugar in mature leaf tissues. The excessive sugar accumulation in GLRaV-3-infected leaves may trigger downstream GLRD symptom development and negatively impact berry quality. We propose a working model to account for the molecular events underlying the pathogenesis of GLRaV-3 and symptom development.

## 1. Introduction

Grapevine (*Vitis* spp.) is a major fruit crop of high economic importance in many countries [1]. According to the latest reports of the International Organization of Vine and Wine, it is estimated that grapes are grown on 7.3 M hectares of land with a total production of over 75 M metric tons worldwide [1,2]. Grapes are mainly used for wine production, reaching 260 M hL globally in 2020 [1]. The produce was also used for consumption as fresh fruit (27.3 MT), and dried grapes (1.3 MT), among other utilities [2]. Unfortunately, grapevines are infected by a large number of viruses, which profoundly limits grape and wine production. Grapevine is host to more than 80 viruses, the largest number of viruses known to infect a single plant species [3,4]. 

Grapevine leafroll disease (GLRD) is economically the most damaging viral disease complex that affects grape and wine production worldwide [5,6]. Vineyards infected with GLRD could suffer yield reductions from 30% to 50% depending on the severity of disease, which negatively impacts the quality of grapes and wine products. It was estimated that GLRD may lead to a lifetime economic loss from $25,000 to $41,000 per hectare over a 25-year span [6,7]. Typical GLRD symptoms include downward curling the mature leaf margins, red to purple discolouration of leaf in dark-berried cultivars, or yellowing of leaves in white-berried cultivars [8]. 

While the etiology of GLRD has yet to be defined, six viral species belonging to three genera of the family *Closteroviridae* are associated with GLRD. These viruses include grapevine leafroll-associated virus 1 (GLRaV-1), GLRaV-3, GLRaV-4, and GLRaV-13 (genus *Ampelovirus*), GLRaV-2 (genus *Closterovirus*), and GLRaV-7 (genus *Velarivirus*). GLRaV-3 is considered the major etiological agent of GLRD and the most destructive grapevine virus due to its global prevalence, consistent association with severe GLRD disease symptoms, and high negative impacts on fruit yield and quality and vineyard lifespan [7,9,10,11,12].

GLRaV-3 is a positive-sense, single-stranded RNA (+ssRNA) virus with one of the largest RNA genomes among plant viruses, only smaller than citrus tristeza virus (CTV, genus *Closterovirus*) and GLRaV-1. The genome of GLRaV-3 varies slightly in size among isolates, ranging from 18.4 to 18.6kb, and encodes 12-13 open reading frames (ORFs) [12]. ORF1a and ORF1b constitute the replication gene block (RGB), which encodes proteins that are involved in genome replication and transcription. ORFs 3-7 make up the quintuple gene block (QGB), which codes for proteins involved in virion assembly and cell-to-cell movement. The remaining ORFs, ORF8–ORF12, are unique to GLRaV-3; their functional roles remain largely unknown. 

The predominant role of GLRaV-3 in GLRD has prompted numerous investigations to elucidate the effect of GLRaV-3 infection on growth, physiology, and fruit quality of the grapevine host (reviewed in [13]). GLRaV-3 infection has been previously linked to dysfunctional photosynthesis in grapevine leaves and altered chemical composition of berries, such as changes in soluble solids content (◦Brix), titratable acidity, pH, and flavonoids, among others [14,15,16,17,18,19,20,21]. However, only limited research is available to our understanding of GLRaV-3-grapevine interactions at the transcriptomic and metabolomic level. Gutha et al. (2010) observed an overall up-regulation of genes involved in the flavonoid biosynthetic pathway in mature leaves of cv. Merlot that was infected with GLRaV-3. In particular, genes involved in anthocyanins biosynthesis were up-regulated, corresponding to the detection of *de novo* anthocyanins synthesis in leaves with GLRD symptoms [22]. Anthocyanins are a group of pigments that give plants the red, purple, or blue colors, which typically accumulate towards the end of a growing season [23]. These findings could, therefore, explain the early onset of reddish–purple discoloration of GLRaV-3-infected grapevine leaves [22].

More detailed analyses of grapevine responses to GLRaV-3 infection were conducted through genome-wide analyses using DNA microarrays. For example, Espinoza et al. (2007a; 2007b) and Vega et al. (2011) analyzed the transcriptomic changes in leaves of GLRaV-3-infected Cabernet Sauvignon and Cabernet Carmenere, and in the berries of Cabernet Sauvignon. These studies reported changes in the expression of genes involved in the biosynthesis of primary and specialized metabolites, cellular communication, cell cycle, as well as defense responses [24,25,26]. It was further revealed that GLRaV-3 infection was associated with delays in berry ripening in cv. Cabernet Sauvignon [26]. Consequently, the delay in berry ripening was associated with the down-regulation of genes involved in sugar metabolism and anthocyanin biosynthesis with a concomitant decline of these metabolites in GLRaV-3-infected berries [26]. 

Though DNA microarray-based transcriptomic studies served as an excellent starting point towards the elucidation of grapevine-GLRaV-3 interactions, they are limited in scope and sensitivity. RNA-Seq has become the method of choice for transcriptome analysis in recent year but has been underutilized in the field of grapevine-virus interactions. Blanco-Ulate et al. (2017) conducted RNA-Seq on berries of a dark-berried cultivar (cv. Zinfandel) that was infected with grapevine red blotch-associated virus (GRBaV, which was later changed to grapevine red blotch virus, GRBV); genes showing perturbed expressional changes were found associated with regulatory pathways involved in berry ripening, including ripening-associated transcription factors, post-transcriptional machinery, and hormonal regulations [27]. More recently, two studies used RNA-Seq to investigate the effects of GLRaV-3 infection on grapevine gene expression. Ghaffari et al. (2020) applied RNA-Seq on berries (cv. Pinot noir) that were either infected with GLRaV-1 or co-infected with GLRaV-1 and GLRaV-3. Their results suggested that GLRD had no direct effect on anthocyanin biosynthesis or sugar transport in the grapevine host [28]. In the second study, Prator et al. (2020) compared the effects of GLRaV-3 infection on the gene expression in grapevine and the experimental host, *Nicotiana benthamiana* that was infected with GLRaV-3 through mealybug transmission. Due to poor quality of the template RNA used and the low output of sequence reads, only a small number of genes were identified as differentially regulated genes (DEG) in both hosts (494 DEGs for grapevine and 157 DEGs for *N. benthamiana*) [29]. Interestingly, the majority of these DEGs were not shared between the two hosts [29]. Results of Prator et al. (2020) suggested the divergent pathogen–host system in GLRaV-3-infected herbaceous plant and woody pant, which should be put into consideration for future experimental design by researchers of similar interest.

Past studies aimed at transcriptomic profiling of the grapevine—GLRaV-3 system focused on a single tissue type (i.e., leaf or berry) or a single developmental stage [24,25,26]. As a perennial woody plant, grapevine exhibits a dynamic relationship between source and sink organs; and this relationship could vary considerably among different developmental stages. A comprehensive profiling of molecular responses of grapevine to GLRaV-3 infection is, therefore, required.

In this study, RNA-Seq was used to study the transcriptomic changes of cv. Cabernet franc infected with GLRaV-3. Gene expressional profiles were generated using both leaf and berry tissues, collected at three defining developmental stages corresponding to young berry, veraison (a stage characterized by the onset of berry ripening when 50% of the berries have changed color [30]), and harvest. Transcriptomic analysis revealed that GLRaV-3 infection altered the expression patterns of the genes involved in photosynthesis, sugar translocation, polyphenolic biosynthesis, and defense response. Interestingly, some of these changes were organ and developmental stage specific. The expressional profiles of a subset of genes were verified by RT-qPCR. Lastly, a working model is presented to account for mechanisms on how GLRaV-3 infection causes GLRD. 

## 2. Materials and Methods

### 2.1. Collection of Grapevine Leaf and Berry Samples

Leaf and berry samples were collected from Cabernet franc clone 210 that was grafted on 3309 and planted in 2001 from a vineyard located in the Niagara Peninsula, Southern Ontario. To identify vines suitable for this study, 35 vines were randomly selected and their viral infection status was tested through multiplex RT-PCR targeting a total of 18 viruses commonly included in the grapevine certification programs in major grape-producing countries. These included five viruses associated with GLRD (GLRaV-1, -2, -3, -4, and -7), four viruses associated with the infectious degeneration and decline (grapevine fanleaf virus/GFLV, Arabis mosaic virus/ArMV, tobacco ringspot virus/TRSV, and tomato ringspot virus/TomRSV), three viruses associated with rugose wood disease complex (grapevine rupestris stem pitting-associated virus/GRSPaV; grapevine virus A/GVA, and grapevine virus B/GVB). In addition, six other viruses, grapevine Syrah virus (GSyV), grapevine red blotch virus (GRBV), grapevine Pinot gris virus (GPGV), grapevine rupestris vein feathering virus (GRVFV), grapevine fleck virus (GFkV), and grapevine asteroid mosaic associated virus (GAMaV) [31]. To ensure consistency in sampling conditions, samples were collected on sunny days within a temperature range that was average for the week in which sampling was to take place. The time for sample collection was set at seven hours after sunrise for all sampling.

Three biological replicates were collected for each experimental group. For example, for the GLRaV-3-positive group, leaves and berries of three vines infected with GLRaV-3 (vine numbers 1B290 1-5, 8-1, 9-2) were collected. Similarly, for the control group, leaf and berry samples were collected from three vines that were free of GLRaV-3 (vine numbers 1B290 5-4, 15-2, and 15-3). These six vines were grown in the same vineyard block and were subjected to identical management practices, such as pest management, trellising, and pruning. 

The developmental stages of grapevine in this study were identified according to the modified Eichhorn and Lorenz system [32,33]. In 2019, leaves were sampled from each of the six vines at E-L 31 (pea-sized berries) and E-L 35 (veraison); while berries were collected at E-L 31, E-L 35, and E-L 38 (harvest) from each of the six vines. In addition, in 2018, leaves at E-L 38 were collected for pilot RNA-Seq run, and were, therefore, not re-collected in 2019.

Shoots located at the lower part of the trunk of each vine facing east were chosen for sampling and were labelled for each subsequent visit. To control for developmental stage-associated variations between biological replicates, leaves with a similar degree of maturity as judged by chlorophyll content were collected. Chlorophyll content index (CCI), which is proportional to the chlorophyll content of leaf was measured by a SPAD meter [34]. SPAD meter reading (non-invasive) of a leaf was taken at five spots that span across the whole area of leaf and average measurement was calculated. The same five spots were measured for every leaf sampled. For each biological replicate, two whole leaves with the highest CCI were collected from the same shoot. Whole berries were collected from the same fruit cluster for each vine at each developmental stage (E-L 31, 35, and 38). Each sample placed in a plastic bag, flash-frozen in liquid nitrogen immediately after sampling, and stored in dry ice for transportation to the lab. Each leaf sample and berry sample were ground to fine powder using a mortar and pestle in the presence of liquid nitrogen, followed by storage at −80 °C until analysis. 

### 2.2. RNA Extraction and Quality Test

Extraction of total RNA of leaves and berries was performed following to the optimized protocol we recently developed [35]. The concentration and purity of all RNA preps as judged by OD260/280 and OD260/230 ratios were assessed by NanoDrop ND-1000 spectrophotometer (Thermo Fisher Scientific, Wilmington, DE, USA). RNA integrity was analyzed through electrophoresis on 1% agarose gel with ethidium bromide staining (5 μL per 100 mL gel). To ensure the suitability of RNA samples for RNA-Seq, RNA integrity of all samples was further assessed by Agilent 2100 Bioanalyzer (RIN > 4), performed by Novogene (Novogene, Sacramento, CA, USA).

### 2.3. RNA-Seq Data Analysis 

RNA preps from leaves and berries (a minimum of 10 μL in volume, 20 ng/μL in concentration per sample) of GLRaV-3-infected and control Cabernet franc vines at different developmental stages (E-L 31 and 35 for leaves; E-L 31, 35, and 38 for berries) were used for mRNA-Seq on NovaSeq 6000 platform to produce 150 bp paired-end reads. RNA-Seq was performed by Novogene (Sacramento, CA, USA). The resulting RNA-Seq data was analyzed using the bioinformatics pipeline we recently developed [13]. False discovery rate (FDR) of <0.05 was commonly used to identify differentially expressed genes in the literature; therefore, the same measurement was used to evaluate genes differentially expressed in GLRaV-3-infected samples. Annotation vocabularies used in functional enrichment analysis were derived from Gene Ontology (GO) [36,37] and were retrieved via Ensemble Plants BioMart (Ensemble Genomes release 51, April 2021) [38]. Functional enrichment analysis was conducted by performing Fisher’s exact test to compare gene lists identified in this study with the non-redundant transcripts of grapevine genome. Multiple test correction was performed using FDR, and a GO term with FDR < 0.05 was considered as enriched.

### 2.4. Identification of Viruses Based on RNA-Seq Data 

RNA-Seq data generated from all samples were first used for detection and confirmation of viruses using Virtool [39]. Virtool is a web-based tool developed by the Canadian Food Inspection Agency (CFIA) that uses NGS data to identify viral sequences present in a sample [39]. Briefly, Virtool (v4.2.1) and its dependencies were installed on a computer following the official Virtool manual (https://docs.Virtool.ca/) (accessed on 27 July 2021). The official reference data for plant viruses (v1.4.0) was imported to Virtool individual account (Hanner lab). Grapevine reference genome (*V. vinifera* cv. PN40024) [40] in FASTA format was up-loaded for Virtool to separate host genome sequences from viral sequences. Raw RNA-Seq data was uploaded to Virtool individual account for identification of presence of virus and viroid sequences.

### 2.5. Quantification of Expression of Select Genes of Interest by RT-qPCR 

For selected genes of interest (GOIs), expressional difference between GLRaV-3-infected and control vines was further examined using RT-qPCR. For reverse transcription, 5 ug of total RNA was used for cDNA synthesis using Applied Biosystem High-Capacity cDNA Reverse Transcription Kit (Thermo Fisher Scientific, Carlsbad, CA, USA) following the instructions provided by the vendor. *CYSTEINE PROTEASE* (CYSP, VIT_03s0038g00280), a gene we recently identified as the most stable gene for RT-qPCR involving grapevine [35], was used as the reference gene for this study. The 13 GOIs examined using RT-qPCR are listed in Table 1.

Primer-BLAST was used to design primers specific for each targeted gene [41]. Primer information is listed in Table 1. RT-qPCR and melt curve analysis were carried out following the protocol as described in Song et al. (2021) with the following modifications: first, each 15 μL reaction contained 5 μL of 10-fold diluted cDNA; and second, RT-qPCR analysis for each sample was performed in duplicate. The amplification efficiency (E) of each gene/primer combination was calculated from a standard curve generated using a five-fold dilution series (1:5 through 1:3125), which was produced via StepOnePlus Software v2.3 (Applied Biosystems, Foster City, CA, USA). The amplification efficiency of the primers used in this study ranged from 84.1% to 99.4% (Table 1). For each GOI, fold change (FC) value of gene expression for samples from vines infected with GLRaV-3, as compared to control samples, was calculated using Pfaffl method [42]. Statistical significance of gene expressional change was evaluated using one-way ANOVA. 

## 3. Results

### 3.1. Selection of Samples and Viral Screening 

To study the interaction between GLRaV-3 and its grapevine host, leaf and berry samples were collected from GLRaV-3-positive and -negative vines (cv. Cabernet franc) grown under the same conditions for RNA extraction and RNA-Seq analysis. Multiplex RT-PCR with primers targeting 18 commonly detected grapevine viruses [31] were used to screen for the presence of GLRaV-3 among other viruses. All the 35 randomly selected Cabernet franc vines tested positive for GRSPaV and GPGV (Appendix A). Therefore, GRSPaV and GPGV were regarded as the ‘background’ viral infections in all grapevine samples used in this study.

RNA-Seq data generated from leaf and berry samples was analyzed via Virtool [39] to further confirm their viral infection status. Results from Virtool analysis were in perfect agreement with those from RT-PCR (Appendix A). Leaf and berry samples of all these six vines tested positive for GRSPaV and GPGV (Appendix A). Virtool analysis confirmed that three vines (1B290 5-4, 15-2 and 15-3) were negative for GLRaV-3 while the other three vines (1B290 1-5, 8-1 and 9-2) were positive for GLRaV-3 (Appendix A). In addition, Virtool revealed the presence of hop stunt viroid and grapevine yellow speckle viroid in the leaf and berry samples, which were known to be commonly detected in wine grapes.

### 3.2. Symptom Development in GLRaV-3-Infected Cabernet Franc 

We observed that some of the leaves of GLRaV-3-infected Cabernet franc vines (located at the base of shoots) developed typical GLRD symptoms at E-L 35 (veraison), while the rest of the leaves on the same vine remained green and lacked GLRD symptoms. The GLRD symptoms included reddish–purple discoloration in the interveinal areas of leaf lamella while the veins remained green (Figure 1b). GLRD symptoms progressed as vines transitioned into E-L 38 stage (harvest), with more leaves on the infected vines developing GLRD symptoms and more intense discoloration of symptomatic leaves. Downward curling of leaf margin was also observed at E-L 38 stage of infected leaves (Figure 1c). No discernible differences were observed between berries from GLRaV-3-infected vs. GLRaV-3-free vines (Figure 1a–c).

### 3.3. Pilot Study in Identification of DEGs Based on RNA-Seq and Transcriptomic Analyses 

In 2018, we collected leaves of control and GLRaV-3-infected Cabernet franc at the E-L 38 stage for a pilot RNA-Seq analysis. Leaf samples at E-L 38 stage were not included in the following RNA-Seq analysis conducted in 2019. Briefly, we identified up-regulation of genes involved in phenylpropanoid and flavonoid biosynthesis in the leaves at E-L 38 of vines infected with GLRaV-3 (Appendix A). On the other hand, processes associated with photosynthesis and carbohydrate metabolism were down-regulated in the same leaves (Appendix A).

### 3.4. RNA-Seq Data Analysis 

An average of 48 million 150-bp paired-end raw sequence reads were generated from each of the 30 samples sequenced via Novogene (leaves at E-L 31 and 35 and berries at E-L 31, 35, and 38, collected from three GLRaV-3-infected vines and three control vines) (Appendix A). After adaptor trimming and quality control, the RNA-Seq dataset from each sample was mapped to the grapevine reference genome (*V. vinifera* cv. PN40024 [40]) via STAR with an average of a 91% uniquely mapped rate (i.e., reads that mapped to only one location on the genome) (Appendix A), indicating high-quality alignment of the RNA-Seq reads. For each experimental group (leaves at E-L 31 and 35; berries at E-L 31, 35, and 38), statistical differences in gene expression between GLRaV-3-infected and control samples were analyzed via DESeq2. As a result, 1469 grapevine genes were significantly up-regulated (up-DEGs) in samples infected with GLRaV-3 (FDR < 0.05). These included 614 up-DEGs for leaf samples (243 up-DEGs at E-L 31 and 371 up-DEGs at E-L 35) and 855 up-DEGs for berry samples (271 up-DEGs at E-L31; 407 up-DEGs at E-L35; and 177 up-DEGs at E-L38) (Appendix A). On the other hand, 1169 genes were significantly down-regulated (down-DEGs) (FDR < 0.05), including 843 down-DEGs for leaf (251 down-DEGs in E-L 31 leaf; 592 down-DEGs in E-L35 leaf) and 326 down-DEGs for berry (59 down-DEGs in E-L31 berry; 246 down-DEGs in E-L35 berry; and 21 down-DEGs in E-L38 berry) (Appendix A). It is of interest to note that the highest number of DEGs were identified at the E-L-35 stage for both leaf and berry samples, suggesting that the most drastic transcriptomic changes happened at veraison in Cabernet franc following GLRaV-3 infection. 

### 3.5. DEGs Associated with Photosynthesis, Carbohydrate Metabolism, and Sugar Transport 

Functional overrepresentation analysis was used to identify key biological processes altered (FDR < 0.05) in grapevine as a consequence of GLRaV-3 infection (Appendix A). For GLRaV-3-infected leaf samples, genes involved in photosynthesis were significantly down-regulated at E-L 31 and E-L 35 stages (Figure 2b; Appendix A). In addition, genes involved in chloroplast rRNA transcription and processing were down-regulated in GLRaV-3-infected leaves at E-L 31; genes involved chlorophyll biosynthetic process were also down-regulated in GLRaV-3-infected leaves at E-L 35 (Figure 2b; Appendix A).

Corresponding with repressed photosynthesis, genes involved in carbohydrate metabolism were down-regulated in GLRaV-3-infected leaves at E-L 35, including those involved in glucogenesis/glycolysis (e.g., *ALDOSE 1-EPIMERASE*), the pentose phosphate pathway (e.g., *TRANSALDOLASE*/VvTALDO), and the Calvin cycle (e.g., *PHOSPHORIBULOKINASE*/VvPRK) (Figure 2b; Appendix A). Genes involved in sucrose and starch metabolism were down-regulated in GLRaV-3-infected leaves, including *SUCROSE SYNTHASE* (VvSUS) and *NEUTRAL INVERTASE* (VvNIV) at E-L 31, and VvNIV and *BETA-AMYLASE* (VvBMY) at E-L 35 (Appendix A). 

Genes involved in cell wall metabolism were differentially regulated in GLRaV-3-infected leaves at E-L 31 and 35. For instance, genes involved in cell wall biogenesis were up-regulated (e.g., *CELLULOSE SYNTHEASE*), while those involved in cell wall softening and expansion were down-regulated (e.g., *EXPANSIN*) (Figure 2; Appendix A).

In accordance with the down-regulated genes involved in photosynthate assimilation, we identified differential regulation of genes involved in sugar transport, particularly for GLRaV-3-infected leaves at E-L 35 stage (Figure 2). For example, genes coding for *PUTATIVE POLYOL/MONOSACCHARIDE TRANSPORTER* (VvPMT3), *HEXOSE TRANSPORTER 3/7* (VvHT3/7), and *SUCROSE TRANSPORTER 2* and *27* (VvSUT2 and VvSUT27) were significantly down-regulated, while genes coding for VvHT2, VvHT5 and *UDP-GALACTOSE TRANSPORTER 3-LIKE* (VvUTR3-like) were significantly up-regulated in GLRaV-3-infected leaves at E-L 35 (FDR < 0.05) (Appendix A). 

Unexpectedly, we observed an opposite trend in GLRaV-3-infected young berries (E-L 31), when compared to leaf samples in the expression patterns of the genes involved in photosynthesis, photosynthates assimilation, and sugar transport. For example, genes involved in photosynthesis, which were down-regulated in GLRaV-3-infected leaf at E-L 31 and E-L 35, were up-regulated in GLRaV-3-infected berries at E-L 31 (Figure 2a). Furthermore, genes involved in carbohydrate metabolism were up-regulated in GLRaV-3-infected berries at E-L 31 (e.g., *PEP CARBOXYLASE*/VvPEPCase) (Figure 2a; Appendix A). Genes involved in cell wall metabolism were up-regulated in GLRaV-3-infected berries at E-L 31 (e.g., *PECTINESTERASE*/VvPE) (Appendix A). VvSUS, a gene involved in sucrose metabolism, was up-regulated in GLRaV-3-infected berry at E-L 35 and E-L 38 stages (Appendix A). As for sugar transport, genes coding for POLYOL TRANSPORTER 5 (VvPLT5), *TRIOSE PHOSPHATE/PHOSPHATE TRANSLOCATOR* (VvTPT), *INOSITOL TRANSPORTER 2* (VvINT2), and *BIDIRECTIONAL SUGAR TRANSPORTER SWEET17* (VvSWEET17) were up-regulated in GLRaV-3-infected berries at E-L 31, and VvSUT27 was up-regulated at E-L 38 (Appendix A). No DEGs associated with sugar transport with FDR < 0.05 were identified in GLRaV-3-infected berry samples at E-L 35. 

Taken together, these results suggested that GLRaV-3 infection has led to repressed expression of genes related to photosynthesis, chloroplast activities, carbohydrate metabolism, and sugar transport in grapevine leaves. Conversely, genes involved in photosynthesis and energy production were up-regulated in GLRaV-3-infected young berries. Genes involved in sucrose metabolism were up-regulated in GLRaV-3-infected berries at veraison and harvest. In addition, GLRaV-3 infection has led to elevated expressions of genes involved in cell wall biogenesis and modification in leaves (E-L 31 and 35) and young berries. 

### 3.6. DEGs Associated with Biosynthesis of Secondary Metabolites 

For leaves, genes related to the biosynthesis of flavonoids and anthocyanin-containing compounds were over-represented among the up-regulated DEGs of GLRaV-3-infected leaves at E-L 35 (Figure 2a; Appendix A), suggesting an up-regulation of the anthocyanin biosynthetic pathway in GLRaV-3-infected leaves at veraison. Such a change was not detected in GLRaV-3-infected leaves at E-L 31. In addition, genes associated with the biosynthesis of flavonol (e.g., *FLAVONOL SYNTHASE/FLAVANONE 3-HYDROXYLASE*/VvFLS) and flavan-3-ols (e.g., *ANTHOCYANIDIN REDUCTASE*/VvANR) were up-regulated in GLRaV-3-infected leaves at E-L 35 (Appendix A). These results suggested an increased capacity to synthesize the three major grapevine flavonoids (anthocyanins, flavonol, and favan-3-ols) in GLRaV-3-infected leaves at veraison. 

Genes involved in the biosynthesis of flavan-3-ols biosynthesis were down-regulated in GLRaV-3-infected berries at E-L 35 (e.g., *LEUCOANTHOCYANIDIN REDUCTASE 1/*VvLAR1), up-regulated at E-L 38 (e.g., VvLAR2), while not altered (FDR < 0.05) at E-L 31 (Appendix A). Interestingly, the phenylpropanoids biosynthetic pathway was significantly enriched (FDR < 0.05) in the up-regulated DEGs of GLRaV-3-infected berries at E-L 38, including those DEGs involved in the synthesis of major flavonoids (e.g., *ANTHOCYANIDIN 3-O-GLUCOSYLTRANSFERASE/*VvUFGT, VvLAR2, and *GLUTATHIONE TRANSFERASE* 3/VvGSTL3) (Appendix A). 

Together, these results suggest an up-regulation of genes involved in biosynthesis of major flavonoids (anthocyanins, flavonol, and flavan-3-ols) in GLRaV-3-infected leaves, while down-regulation of flavan-3-ols in GLRaV-3-infected berries at veraison. On the other hand, genes involved in the biosynthesis of major flavonoids were up-regulated in GLRaV-3-infected berries at harvest. 

### 3.7. DEGs Involved in Mitochondrial Activities 

We identified several down-regulated genes in GLRaV-3-infected leaves at E-L 35 that were associated with maintaining the structure and function of mitochondria, including genes coding for mitochondrial ribosomal proteins, translation elongation factors, electron transfer flavoprotein components, and glutamate dehydrogenase (Appendix A). 

Similar to the GLRaV-3-infected leaves at E-L 35, we also identified down-regulation of genes associated with mitochondrial activity in berry samples collected at E-L 35 (Appendix A). Genes associated with protein insertion into the mitochondrial inner membrane (*MITOCHONDRIAL IMPORT INNER MEMBRANE TRANSLOCASE SUBUNIT TIM13* /VvTIM13, *REACTIVE OXYGEN SPECIES MODULATOR 1*/VvROMO1), and genes encoding mitochondrial respiratory chain complex I (VIT_01s0137g00340, VIT_05s0020g02150, VIT_06s0080g00540, and VIT_11s0103g00270) were down-regulated in GLRaV-3-infected berries at E-L 35 (Appendix A). It is worth noting that GLRaV-3 was reported to form viral replication complexes (VRC) in association with mitochondrial outer membrane of infected cells [43,44,45]. The repressed expression of genes related to mitochondrial function could be indicative of organelle damage caused by GLRaV-3 infection. 

### 3.8. DEGs Associated with Defense Responses: PR Proteins, R Genes, and RNA Silencing

We identified a range of up-regulated genes involved in host defense responses in both leaves and berries of GLRaV3-infected vines (Figure 2a). For example, members of *PATHOGENESIS-RELATED* (PR) genes were up-regulated. These included the up-regulation of VvPR-2 and VvPR-10.3 in leaf samples at E-L 31; VvPR-2, VvPR-4A, VvPR-5, and VvPR-10.3 in leaf samples at E-L 35; VvPR-2, VvPR-5, and PR-10.3 in berry samples at E-L 35; and VvPR-2 and VvPR-10.3 in berry samples at E-L 38 (Appendix A). Interestingly, VvPR-10.7 was down-regulated in leaf samples at E-L 35 and berry samples at E-L 38 (Appendix A).

We also found up-regulation of several *RESISTANCE* (R) genes at the E-L 35 stage in both leaves and berries of vines infected with GLRaV-3. These up-regulated R genes code for proteins of the NUCLEOTIDE-BINDING SITE LEUCINE-RICH REPEAT (NB-LRR) family of R proteins, which are involved in effector-triggered immunity (ETI), including TIR domain-containing protein, NB-ARC domain-containing protein, and RN_N domain-containing protein (Appendix A). These R proteins are important in the recognition of invading pathogens such as viruses [46]. 

Several genes involved in RNA silencing were up-regulated in GLRaV-3-infected berries at E-L 35, including *RNA-DEPENDENT RNA POLYMERASE 1* (VvRdRp1), *RNA-DEPENDENT RNA POLYMERASE 6* (VvRdRp6), *ENDORIBONUCLEASE DICER 2* (VvDcr2), *DICER-LIKE PROTEIN 4* (VvDCL4), *PROTEIN ARGONAUTE 4A* (VvAGO4A), *PROTEIN ARGONAUTE 5* (VvAGO5), and RNA-DIRECTED DNA METHYLATION 3 (VvRDM3) (Appendix A). DEGs associated with RNA silencing were not identified in other stages of berry or any stages of leaf samples with the FDR <0.05 threshold. 

### 3.9. DEGs Associated with Stress Response, Senescence and Hormonal Signaling

A large number of genes encoding proteins that belong to the general *HEAT SHOCK PROTEIN* family (HSP) and *HSP CO-CHAPERONS* were up-regulated in GLRaV-3-infected leaves at E-L 31 and 35 and berries at E-L 35 and 38. These up-regulated genes encode small HSPs, DNAJ/HSP40, HSP70s, HSP90s, HSP90 activator, and BAG family of molecular chaperone regulators (BAGs) (Appendix A). HSPs and their associated co-factors are stress-responsive proteins that have a multitude of functions such as folding newly synthesized proteins, refolding proteins that are denatured, stabilizing proteins that have unstable structures due to stressful events, and intracellular protein transport [47,48].

We also revealed enhanced expression of a large number of genes encoding *ETHYLENE-RESPONSIVE TRANSCRIPTION FACTORS* (ERFs) and WRKY transcription factors in leaves at E-L 31 and E-L 35 and berries at E-L 38 of vines infected with GLRaV-3 (Appendix A). Interestingly, all of the ERFs were down-regulated in GLRaV-3-infected berries at E-L 35. These changes were not identified in berry samples at E-L 31. These transcription regulators play key roles in activating the transcription of many genes associated with both biotic and abiotic stress, senescence, seed dormancy, or the regulation of phytohormone signaling pathways [reviewed by [49,50]].

### 3.10. RT-qPCR Validation of Selected DEGs of Interest

To validate results of transcriptomics analysis, we selected and examined the expressional profile of 13 DEGs through RT-qPCR. These DEGs were selected to represent each category of genes whose expression was shown to be affected by GLRaV-3 infection based on RNA-Seq analysis as discussed above. These included one DEG involved in photosynthesis (*PROBABLE OXYGEN-EVOLVING ENHANCER PROTEIN 2*/VvPSBP1), four DEGs involved in carbohydrate partitioning (VvSUT2, VvHT5, *GRANULE-BOUND STARCH SYNTHASE 1/*VvGBSS1, and VvSUS), three DEGs involved in the biosynthesis of flavonoids (VvUFGT, VvLAR1, and VvFLS), two involved in defense against pathogens (VvPR-10.7, and VvPR-10.3), and three involved in mitochondrial activities (*ELECTRON TRANSFER FLAVOPROTEIN SUBUNIT BETA*/VvETFB, VvTIM13, *NADH DEHYDROGENASE (UBIQUINONE) 1 ALPHA SUBCOMPLEX SUBUNIT 1/*VvNDUFA1) (Table 1). In general, results from RT-qPCR confirmed those derived from transcriptomic analysis (Figure 3). As reported by others [51], minor ‘non-concordance’ between RNA-Seq and RT-qPCR results could be found in those genes showing fold change lower than 2 of RT-qPCR results. In the present study, these genes included VvSUS in berry at E-L 35 and VvUFGT in berry at E-L 38. Below we describe in detail the RT-qPCR results of these DEGs based on their functional category.

### 3.11. DEGs Involved in Photosynthesis, Energy Production, and Sugar Transport

The VvPSBP1 gene likely codes for a subunit of the photosystem II (PSII) supercomplex [52]. RT-qPCR results showed that VvPSBP1 gene was down-regulated in GLRaV-3-infected leaves at E-L 31 (by 1.48-fold) and 35 (2.32-fold), while up-regulated in berries at E-L 31 (1.24-fold) (Figure 3a: PSBP1 chart). This result corroborates findings from the transcriptomic analysis (Appendix A).

Four genes involved in carbohydrate partitioning were validated via RT-qPCR. VvSUT2, a H^+^/sucrose symporter, was significantly down-regulated in GLRaV-3-infected leaves at E-L 31 (1.32-fold) and E-L 35 (2.63-fold), while up-regulated in berries at E-L 31 (2.44-fold), E-L 35 (3.49-fold), and E-L 38 (1.29-fold) (Figure 3b: SUT2 chart). VvHT5, a H^+^/monosaccharide symporter, was significantly up-regulated in GLRaV-3-infected leaves at E-L 31 (2.39-fold) and E-L 35 (3.86-fold), while down-regulated in GLRaV-3-infected berries at E-L 35 (1.32-fold) and E-L 38 (1.33-fold) (Figure 3b: HT5 chart). VvSUS, believed to be involved in the reversible cleavage of sucrose, was significantly down-regulated in GLRaV-3-infected leaves at E-L 31 (5.09-fold) and E-L 35 (2.21-fold), while up-regulated in berries at E-L 31 (1.52-fold) and E-L 38 (13.54-fold) (Figure 3b: SUS chart). For GLRaV-3-infected berries at E-L 35, VvSUS was down-regulated as judged by RT-qPCR (1.51-fold), while up-regulated based on transcriptomic analysis (3.57-fold) (Appendix A). The expression of VvGBSS1 gene, which the product is centrally involved in starch synthesis, was significantly up-regulated in GLRaV-3-infected leaves at E-L 31 (1.54-fold), E-L 35 (3.91-fold), and berries at E-L 31 (1.23-fold), while significantly down-regulated in berries at E-L 38 (1.45-fold) (Figure 3b: GBSS1 chart). 

### 3.12. DEGs Involved in the Biosynthesis of Major Flavonoids

Three genes encoding key enzymes involved in flavonoid biosynthesis were validated via RT-qPCR. UFGT, LAR1, and FLS are involved in biosynthesis of the three major flavonoids of grapevine: anthocyanins, flavan-3-ols, and flavonols, respectively. The enzyme UFGT catalyzes the conversion of anthocyanidins to anthocyanins, which are responsible for the pigmentation of leaves and berries starting around veraison [53,54]. LAR1 catalyzes the synthesis of 2,3-trans-flavan-3-ols monomers, including catechin and gallocatechin [54,55]. FLS is the key enzyme that catalyzes biosynthesis of flavonols, including kaempferol, quercetin, and myricetin [53,54].

RT-qPCR analysis showed that the VvUFGT gene was significantly up-regulated in GLRaV-3-infected leaves at E-L 31 (4.47-fold), E-L 35 (1.44-fold), and berries at E-L 31 (1.98-fold) (Figure 3a: UFGT chart). For GLRaV-3-infected berries at E-L 38, VvUFGT was significantly down-regulated as shown by RT-qPCR (1.68-fold), while significantly up-regulated based on transcriptomic analysis (1.84-fold) (Appendix A). VvLAR1 was significantly up-regulated in GLRaV-3-infected leaves at E-L 35 (5.32-fold), while significantly down-regulated in berries at E-L 31 (1.43-fold) (Figure 3a: LAR1 chart). VvFLS had a significant up-regulation in GLRaV-3-infected leaves at E-L 31 (1.53-fold), E-L 35 (13.25-fold), and in GLRaV-3-infected berries at E-L 31 (1.35-fold) and E-L 35 (2.55-fold). VvFLS was down-regulated in berries from GLRaV-3-infected vines at E-L 38 (2.16-fold) (Figure 3a, FLS chart). 

### 3.13. DEGs Involved in Defense against Pathogens

Two genes each encoding an isoform of PR-10 protein family members (PR-10.7 and PR-10.3) were selected to represent this category of genes via RT-qPCR analysis. VvPR-10.3 and VvPR-10.7 were regulated in an opposite pattern in GLRaV-3-infected leaf samples. For example, VvPR-10.3 was up-regulated (4.3-fold), while VvPR-10.7 was down-regulated (2.15-fold) in GLRaV-3-infected leaves at E-L 35 (Figure 3c: PR-10.3 and PR-10.7 charts). For berry samples, both VvPR-10.3 and VvPR-10.7 were significantly down-regulated in GLRaV-3-infected berries at E-L 31, and up-regulated at E-L35. VvPR10.7 was significantly down-regulated (2.01-fold) in berry at E-L 38 (Figure 3c: PR-10.3 and PR-10.7 charts). These results are in general agreement with those from transcriptomic results (Appendix A). 

### 3.14. DEGs Involved in Mitochondrial Activities 

One gene involved in maintaining mitochondrial structure (VvETFB), and two genes involved in mitochondrial activities (VvTIM13, and VvNDUFA1) were validated via RT-qPCR. Overall, all three genes were significantly down-regulated in leaves of vines infected with GLRaV-3. For example, VvETFB was down-regulated by 1.31-fold and 1.63-fold, while VvTIM13 was down-regulated by 1.18-fold and 1.55-fold in GLRaV-3-infected leaves at E-L 31 and E-L 35, respectively (Figure 3d: ETFB and TIM13 chart). VvNDUFA1 was down-regulated by 1.22-fold in GLRaV-3-infected leaves at E-L 31 (Figure 3d: NDUFA1 chart). Consistent with results of transcriptomic analysis (Appendix A), VvTIM13 and VvNDUFA1 were also significantly down-regulated in berries at E-L 35 as a result of GLRaV-3 infection (Figure 3d: TIM13 and NDUFA1 chart).

## 4. Discussion

GLRaV-3, a member of the genus *Ampelovirus* (family *Closteroviridae*), is the chief etiological agent associated with GLRD [11,12,56]. Unfortunately, interactions between GLRaV-3 and grapevine host at the molecular, cellular, and physiological level have not been extensively studied. This lack of understanding is attributable to various factors that are unique to woody perennials, such as grapevine. For example, grapevine is commonly mix-infected with multiple viruses, making it very difficult to sort out the impact of a given virus on the host. Mixed infections would further complicate the matter as they could lead to greater damage compared to single infection as a result of synergistic effect [57]. 

This study was initiated to unravel the impacts of GLRaV-3 infection on the global gene expression of the grapevine host to understand the pathogenesis of GLRaV-3. Two types of tissues (leaf and berry) collected at early (E-L 31), middle (E-L 35), and late (E-L 38) stages of grapevine development were subjected to RNA-Seq to identify differentially expressed genes. Though the identified DEGs were distributed among diverse biological processes and pathways, a major portion of them were associated with photosynthesis, sugar transport, polyphenolic synthesis, stress response, and defense as well as mitochondrial activities. To extract key information out of the overwhelmingly large datasets, we focus the discussion on several key aspects pertaining to the pathogenesis of GLRD. A working model is proposed to explain possible mechanisms of GLRaV-3 infection leading to GLRD (Figure 4).

### 4.1. GLRaV-3 Infection Alters Expression of Genes Involved in Source–Sink Relationship, Sugar Transport, and Carbohydrate Metabolism 

An altered carbohydrate metabolism and source–sink relationship of grapevine leaves and young berries were identified at transcriptomic level due to GLRaV-3 infection. Leaves of Cabernet franc infected with GLRaV-3 at both E-L 31 and E-L 35 showed repression of genes related to photosynthesis, chlorophyll biosynthesis, carbohydrate metabolism (glycolysis/glucogenesis, pentose phosphate pathway, and Calvin cycle), while up-regulation of genes involved in cell wall biosynthesis and remodeling (Figure 2). These results were in-line with earlier studies. For example, genes encoding photosynthetic proteins and chlorophyll biosynthesis enzymes were shown to be down-regulated in leaves of GLRaV-3-infected Cabernet Sauvignon and Cabernet Carmenere [24]. These results were in agreement with GLRaV-3-associated changes found at the metabolites level. Several studies revealed that grapevine leaves had reduced chlorophyll and carotenoid content, lower net CO^2^ assimilation, and decreased photosynthetic efficiency as a result of GLRaV-3 infection [16,20,58,59,60,61,62,63,64]. Cell wall reinforcement and remodeling were considered one of the basal defense responses of plants against biotic or abiotic stresses [65,66,67,68]. The up-regulation of genes involved in cell wall biosynthesis may be an indication of an activated general defense triggered by GLRaV-3 infection, manifesting as an alteration of carbohydrate partitioning.

### 4.2. Impairment of Sugar Export from Source Leaves

The present study showed that GLRaV-3 infection led to the down-regulation of genes encoding sucrose transporters (VvSUT2 and VvSUT27) in leaves (Figure 3b, Appendix A). VvSUT2 and VvSTU27 are paralogs of each other [38], and likely have similar functions. Homologs of VvSUT2 and VvSUT27 include SlSUT1 of tomato and AtSUC2 of *A. thaliana* [69,70]. Functional characterization showed that AtSUC2 is a H^+^ symporter localized in the plasma membrane of companion cells (CC), and is responsible for the importation of sucrose from the apoplast into the cytoplasm of CC [71,72]. On the other hand, SlSUT1 knockout tomato plants had an increased accumulation of soluble sugars in leaves, suggesting that SlSUT1 may play a crucial role in loading sugar from source tissue to the phloem [69]. It is logical to speculate that VvSUT2 and VvSTU27 may be localized to CC and function to upload sucrose from photosynthetic cells into the phloem. The repression of VvSUT2 and VvSTU27 gene expression in GLRaV-3-infected leaves may suggest impairment in sugar transport from the apoplast into the phloem of leaf tissue. 

In accordance with the down-regulation of genes involved in sugar export from GLRaV-3-infected leaf tissues, we identified increased expression of VvHT5 gene in leaves infected with GLRaV-3, prompting speculation of an increased activity in hexose retrieval back to parenchyma cells. VvHT5 is likely localized to plasma membrane and is responsible for transporting glucose and fructose [73,74]. AtSTP13 of *A. thaliana*, a VvHT5 homolog [73], was characterized as a high affinity hexose symporter located on plasma membrane and involved in retrieving both fructose and glucose from the apoplast to the cytoplasm of photosynthetic cells [75]. VvHT5 likely has a role similar to AtSTP13 in retrieving hexoses from apoplast. Up-regulation of VvHT5 gene can be construed as a response of grapevine leaf tissue to the accumulation of excess sucrose in leaf apoplast due to the down-regulation of VvSUTs. Such a speculation may be further examined at enzymatic and metabolic level. It is expected that sucrose could not be efficiently up-loaded to the phloem of GLRaV-3-infected vines due to the repressed activity of VvSUTs (VvSUT2 and 27). Consequently, excess sugar in the apoplast of GLRaV-3-infected leaf could be hydrolyzed to hexoses, possibly by acidic cell-wall bound invertase. In turn, excess hexose in the apoplast would be taken back into leaf parenchyma cells via up-regulated VvHT5. Interestingly, up-regulation of VvHT5 gene was also identified in grapevine leaves infected with powdery and downy mildew or subjected to wounding [73]. These observations suggested a role for VvHT5 in grapevine general defense against stress, possibly through increased sugar retrieval to supply energy needed by the leaf tissue during stress [73].

This study identified the up-regulation of VvHT2 and down-regulation of VvHT3/7 in GLRaV-3-infected leaf (Appendix A). VvHT2 is a grape homolog of AtSTP5, while VvHT3/7 is a homolog of AtSTP7 [70]. While both AtSTP5 and AtSTP7 were predicted to be anchored in the plasma membrane, neither of them showed monosaccharide transporter activity through Baker’s yeast assay where various substrates, including glucose and fructose, were tested [75]. It was reasoned that VvHT2 and VvHT3/7 do not have hexose transporter activity either. It is puzzling why the expression of VvHT2 was up-regulated while VvHT3/7 was down-regulated in GLRaV-3-infected leaves at E-L 35. It is possible that these two VvHTs may have other unknown functions which were differentially regulated during GLRaV-3 infection. Further work is needed to understand their roles in grapevine response to GLRaV-3.

### 4.3. Increased Expression of Genes in Photosynthesis and Related Activities in Berries as a Compensatory Mechanism in Response to Shortage in Sugar Supply

The identification of up-regulation of genes involved in photosynthesis and carbohydrate metabolism in leaves and an opposite pattern in young berries of GLRaV-3-infected Cabernet franc suggested a change in source–sink dynamic between GLRaV-3-infected leaves and young berries (Figure 2; Appendix A). We speculate that, as a mechanism to compensate for the shortage in sucrose supply from leaves due to GLRaV-3-infection, young berries attempted to increase their capacity for photosynthesis, carbohydrate metabolism, and sugar transporter activities.

For berries at veraison and harvest, the expression of both VvSUT2 and VvSUT27 was up-regulated in GLRaV-3-infected vines (Appendix A; Figure 3b: SUT2 chart). Starting from veraison, berries rely on the import of carbohydrates from photosynthetic tissue via the apoplastic route [76]. This kind of sugar import would likely involve the participation of sugar transporters, such as VvSUTs. It was suggested that SUTs may function in sucrose downloading when expressed in sink tissues [72]. Based on these reasons, we expected that the expression of VvSUTs in infected berries would decrease because of the shortage in sugar supply from leaf tissue. It is, therefore, surprising to see an up-regulation of VvSUTs in GLRaV-3-infected berries at veraison and harvest. However, it remains uncertain as to the functions of VvSUTs in grape berries and if the up-regulation of VvSUTs genes signifies increased sugar import, leading to increased sugar accumulation in GLRaV-3-infected berries. The best way to validate is to directly measure soluble sugar levels in GLRaV-3-infected berries. Furthermore, the localization and functions of VvSUTs in berries also need to be characterized. 

### 4.4. GLRaV-3 Infection Affects Expression of Genes Involved in Polyphenolic Biosynthesis Due to Altered Source–Sink Relationship 

The induction of anthocyanin biosynthesis was tightly associated with increased sugar levels [77,78]. Based on the transcriptomic results from this study, it is hypothesized that sugar export from leaves was impaired due to GLRaV-3 infection, which, in turn, would lead to sugar accumulation in leaves. Sugar accumulation would alter carbohydrates partitioning, leading to biosynthesis and accumulation of anthocyanins in GLRaV-3-infected leaves. The induction of anthocyanin biosynthesis was supported by the up-regulation of key genes involved in anthocyanin biosynthetic pathway in GLRaV-3-infected leaves at E-L 35 (Figure 3a; Appendix A). These results were also consistent with the observed red-purple discoloration symptoms in leaves (Figure 1b). Our results were in agreement with Gutha et al. (2010) in that the red-purple pigmentation of leaves in dark-berried cultivars was a result of increased anthocyanin synthesis [22].

In addition to anthocyanins, genes involved in the biosynthesis of flavan-3-ols and flavonols were also up-regulated in GLRaV-3-infected leaves (Figure 3a; Appendix A). This finding was consistent with Gutha et al. (2010), in which up-regulation of genes associated with flavan-3-ols and flavonols biosynthesis as well as increased accumulation of flavan-3-ols and flavonols at metabolic level were found in GLRaV-3-infected leaves [22]. Flavan-3-ols are long recognized as phytoalexins synthesized by the host plant in defense against pathogens, insects, or herbivores [79]. Flavonols, on the other hand, play a role in protecting plant from UV damage [80]. 

### 4.5. GLRaV-3 Infection Induces the Expression of Genes Involved in Pathogen-Targeted Defense 

Several genes coding for R proteins were up-regulated. R proteins are involved in plant defense against pathogens through intercepting pathogen-produced proteins called *effectors* [81,82]. Such a recognition induces ETI and, in turn, leads to the transcriptional activation of defense-related genes, MAPK signaling, the production of signaling molecules such as ROS, nitrogen oxide (NO), Ca^2+^, and phytohormones [81,83,84]. R proteins recognize pathogens either by directly binding pathogen effectors or by forming multiprotein-recognition complexes, possibly with additional host proteins, such as kinases [81,84]. It is possible that the R proteins identified in this study play important roles in the formation of host recognition machinery specifically for GLRaV-3. There has been no research on this subject. Exploring the involvement of the R proteins identified here in grapevine defense against infection by GLRaV-3 and other viruses associated with GLRD may prove to be an interesting topic of future research. 

RNA silencing is a mechanism used by plants to defend against viral infections via homology-dependent degradation of viral RNAs. This study identified a number of up-regulated genes likely involved in RNA silencing against GLRaV-3 infection (VvRdRp1, VvRdRp6, VvDcr2, VvDCL4, VvAGO4A, VvAGO5, and VvRDM3) (Appendix A). For example, Dcr2 and DCL4 are nucleases that process dsRNAs into small interfering RNAs (siRNAs), which were then used as guide for viral RNAs recognition [85]. The protein encoded by these RNA silencing genes may act as the direct contact point between grapevine and GLRaV-3, which may be subjected to counter-defense by the virus. Pathogen effectors are able to evade host detection and inhibit host signaling and defense responses [86,87]. In particular, some viral effectors are able to inhibit host RNA silencing mechanism, therefore recognized as RNA silencing suppressor [88]. It is interesting to note that the protein encoded by ORF10 of GLRaV-3 showed activity of RNA silencing suppressor [89]. How GLRaV-3 may interact with grapevine RNA silencing mechanism remains to be an interesting topic for further study.

### 4.6. A Working Model on GLRaV-3 and Grapevine Interaction 

As a first step toward the elucidation of GLRaV-3 pathogenesis, this paper presents a rudimentary model below (Figure 4). Refinement will certainly be necessary as more information becomes available. GLRaV-3 is a phloem-limited virus believed to replicate primarily in phloem CCs, wherein it is believed to form viral replication complexes on the outer membrane of mitochondria [44,45]. Continued viral replication may induce damage of mitochondrial structure and function. In turn, this would lead to repressed ATP synthesis in GLRaV-3-infected cells. Indirect evidence in support of damaged mitochondria is obtained from the present study, where down-regulation of genes involved in mitochondrial activities in GLRaV-3-infected leaf and berry tissues were identified.

VvSUTs are sucrose/H^+^ symporters located at the plasma membrane of CC. These VvSUTs translocate sugar from the apoplast into the cytoplasm of CC. This process is driven by a proton gradient that is maintained by ATP hydrolysis. Insufficient supply of ATPs, therefore, would lead to repressed VvSUTs activity. As a result, sucrose could excessively accumulate in the apoplast of grapevine leaves, which triggered breakdown of sucrose into hexoses and importation of hexoses from apoplast to cytoplasm of parenchyma cells via up-regulated VvHTs activity. Additionally, PD blockage may be activated as a mean to prevent GLRaV-3 spread (Figure 4), further impeding phloem loading. Excess sugar in mesophyll cell could repress photosynthesis via negative feedback [90], which in turn, would trigger the inhibition of chlorophyll synthesis on the one hand and chlorophyll degradation on the other. Reduction in chlorophyll, and the lack of anthocyanin biosynthetic activity would explain the early onset of yellowing and chlorosis exhibited in leaves of white-berried cultivars as a result of GLRD. Over-accumulation of sugar inside mesophyll cells would prompt alternative carbohydrates partitioning to other biosynthetic pathways, such as cell wall strengthening and anthocyanins biosynthesis. The *de novo* anthocyanins biosynthesis would lead to the red-purple discoloration of GLRD-symptomatic leaves of dark-berried cultivars (Figure 4). 

Excess sugar levels in leaf cells increase osmotic pressure and cause cell expansion. As the parenchyma cells in the palisade layer are densely packed, their expansion would cause palisade layer to ‘push over’ the sponge mesophyll, manifesting in the downward curling of mature leaves observed in both white- and dark-berried cultivars (Figure 4).

Lastly, GLRaV-3 infection alters the source and sink relationship. As photosynthates also play important roles in supporting other sink tissues, such as shoot tips, flower buds, and roots, compromised sugar supply would inevitably slow down grapevine growth, development, and yield as disease progresses. The cumulative effect would render grapevines more susceptible to winter injury under cool climate conditions and shorten the productive lifespan of a vineyard. 

## 5. Conclusions and Future Research

To our best knowledge, this research represents the first attempt to unravel mechanisms that underpin grapevine-GLRaV-3 interactions leading to GLRD through RNA-Seq and global transcriptome analysis. Our study identified alterations in the expression of genes involved in several key biological processes including photosynthesis, sugar transport, carbohydrate metabolism, anthocyanin biosynthesis as well as defense response. These findings prompted us to conceptualize a working model to account for the pathogenesis of GLRD. This working model could serve as a trailblazer for grapevine virology and an inspiration for others in joining a collective effort to understand these highly complex grapevine–virus interactions. Of course, this is only a starting point. Many questions need to be answered. Some of the immediate future research would include the validation of DEGs through metabolomics, mechanisms of mitochondria targeting by GLRaV-3 and the biogenesis of VRCs in association with mitochondrial outer membrane, host defense against infections by GLRaV-3 and other viruses associated with GLRD. Furthermore, the cumulative effects of GLRaV-3 infection on the grapevine host over the lifespan warrant investigation. Lastly, the synergistic effects of distinct viruses that co-infect grapevine are also important area of investigation.

We do recognize and would like to point out that background viral infection involving GRPaV and GPGV occurred in all Cabernet franc samples collected. As a result, the presence of these two viruses could potentially interfere with GLRaV-3, creating possibilities for either antagonistic or synergistic interactions, and, thus, skewing some of the data on host gene expression due to GLRaV-3, as well as some of the conclusions. The use of virus-free propagation stocks and infectious clone for GLRaV-3 would offer the ultimate condition to test the effects of singular infection with GLRaV-3 on grapevine gene expression.

## Figures and Tables

**Figure 1 viruses-14-01831-f001:**
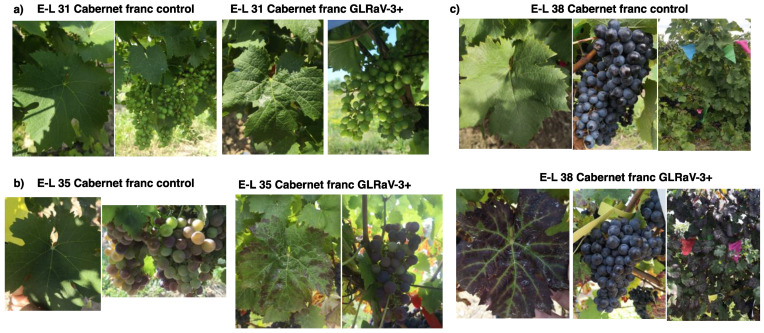
Symptom progression of grapevine leafroll-associated virus 3 (GLRaV-3) infection in leaves and berries of Cabernet franc. The three developmental stages of grapevine were identified based on the modified Eichhorn and Lorenz system (Coombe, 1995; Dry and Coombe, 2004). (**a**): E-L31: pea-sized berry (photographs taken on 23 July 2019); (**b**) E-L 35: veraison (11 September 2019); (**c**) E-L 38: harvest (11 October 2019).

**Figure 2 viruses-14-01831-f002:**
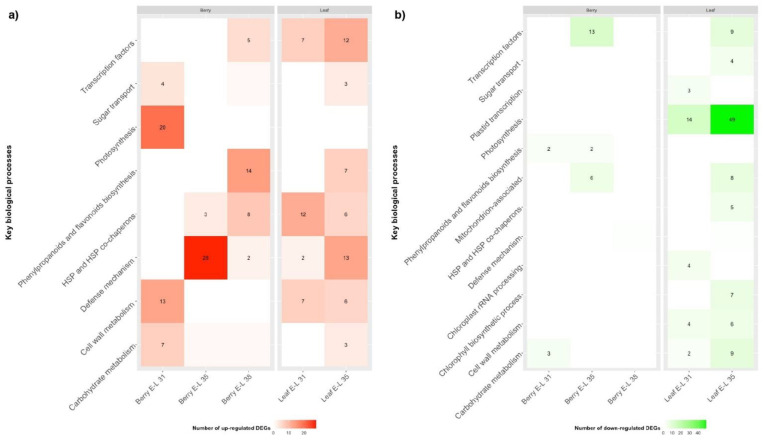
Heatmap of key biological processes up-regulated (**a**) or down-regulated (**b**) in GLRaV-3-infected leaf and berry at E-L 31, 35, and 38 developmental stages. Red colour indicates biological process being up-regulated, green colour indicates down-regulation. Color intensities correlate with the number of DEGs identified in each biological process. Biological processes (square spaces) with DEG number equal or smaller than 1 were not numbered. The complete list of enriched biological processes (*p* < 0.05) can be accessed in Appendix A.

**Figure 3 viruses-14-01831-f003:**
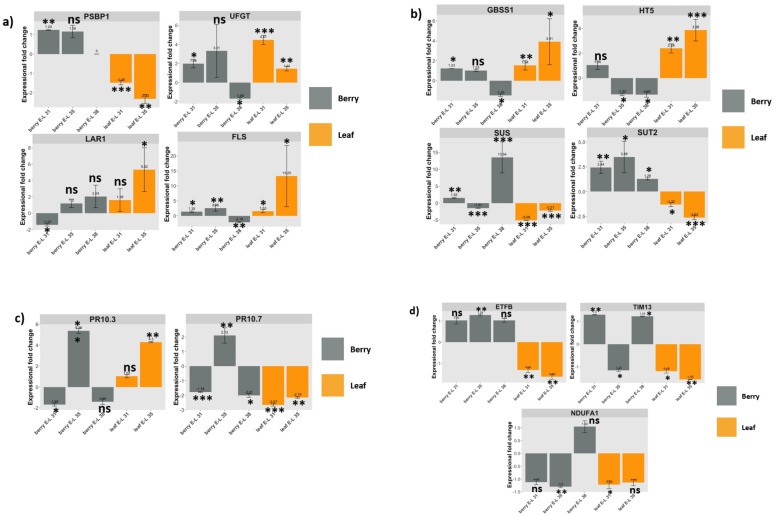
Expression patters of 12 genes of interest (GOIs) measured by RT-qPCR. Fold change of each GOI was compared between GLRaV-3-infected samples and control samples in leaf (orange bars) at E-L 31 and E-L 35 stage and in berry (grey bars) at E-L 31, 35, and 38 stages. (**a**): Fold change of genes representing photosynthesis (PSBP) and flavonoid biosynthesis (UFGT, LAR1, and FLS); (**b**): Fold change of genes involved in carbohydrate partitioning (GBSS1, HT5, SUS, and SUT2); (**c**): Fold change of two genes involved in defense response against pathogens (PR10.3 and PR10.7); (**d**): Fold change of three genes involved in mitochondrial activities (ETFB, TIM13, NDUFA1) in leaf and berry samples at the respective developmental stages when infected with GLRaV-3. * Fold change of VvPSBP1 was not examined in berries at E-L 38 and the value was shown as 0. Relative expression values are mean, ± SD (error bars), *n* = 3. ns: not significant (*p* > 0.05), meaning there is no significant change in gene expression identified for the GOI in GLRaV-3-infected sample compared to GLRaV-3-free sample. * significant (0.01 < *p* < 0.05); ** significant (0.001 < *p* <0.01); and *** significant (*p* < 0.001).

**Figure 4 viruses-14-01831-f004:**
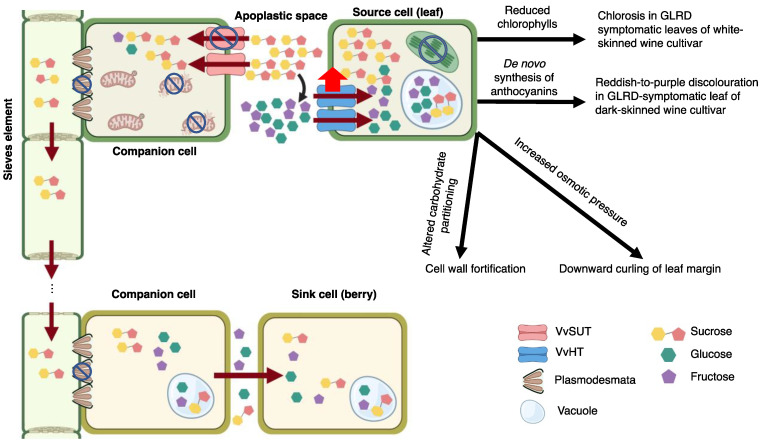
Working model on GLRaV-3-grapevine interactions leading to GLRD. GLRaV-3 replicates in association with the outer membrane of mitochondria, leading to mitochondrial damage and repressed ATP generation. VvSUT transports sucrose in an ATP-dependent manner. Loss of ATP in companion cells (CC) leads to down-regulated VvSUT activity, thereby, decreased sucrose import from apoplast to cytoplasm of CC. Increased sucrose accumulation in apoplast leads to increased hydrolysis of sucrose into glucose and fructose by acidic cell wall invertase (curved black arrow). Excess hexoses are then retrieved back into cells of source leaves via up-regulated VvHT. Consequently, photosynthesis in source cells was repressed as a result of negative feedback from excess sugar in source cells. This in turn leads to a decrease in chlorophyll biosynthesis and increase in chlorophyll degradation. Reduced chlorophylls lead to early onset of chlorosis in leaves of white-berried cultivars. Sugar accumulation in source cells leads to increased osmotic pressure, inducing downward curling of leaf margin. In addition, excessive sugar triggers alternative carbohydrate partitioning, including cell wall biosynthesis and biosynthesis of secondary metabolites, such as anthocyanins. *De novo* biosynthesis of anthocyanins leads to reddish-to-purple discolouration in GLRD-symptomatic leaf of dark-berried cultivar. GLRaV-3 infection may also trigger callose deposition around plasmodesmata (PD), causing further blockage of sugar translocation between source and sink tissues. The net outcome is reduced uptake of sugar into the phloem and blockage of sugar translocation to the sink organ (berries), leading to smaller size and delayed maturation of a portion of the berries in a cluster. The degree of impact on berry cluster depends largely on the viral titer and the damage afflicted on the mitochondria. Note: cell organelles not discussed in the working model were omitted in the illustration figure. Created with https://BioRender.com (accessed on 10 March 2022). Publication license has been obtained.

**Table 1 viruses-14-01831-t001:** Primer design information.

Genes	EnsemblPlant Gene ID	NCBI Accession No.	NCBI Gene Description	Primer Sequence (5′-3′)	Amplicon Size (bp)	Amplification Efficiency (E%)
VvCYSP	VIT_03s0038g00280	NM_001281060.1	Cysteine protease	F:AAAATCAGGGTTCGTGTGGGTC	190	96.791
				R:GCAGTGTTCATCAGCCCACC		
VvUFGT	VIT_12s0034g00130	XM_010659535.2	Anthocyanidin 3-O-glucosyltransferase 2	F:TCTTCCCTTCTGTGGTGCTTG	187	99.108
				R:TTATTGAGCAGGGGTCCAACAG		
VvLAR1	VIT_01s0011g02960	NM_001280958.1	Leucoanthocyanidin reductase 1	F:AACAGTGGACGATGTCCGAAC	178	86.137
				R:CTGTGGGATGATGTTTTCTCCG		
VvFLS	VIT_18s0001g03430	XM_002285805.3	Flavonol synthase/flavanone 3-hydroxylase	F:ATGCCCTCTTTGTCCATGTC	190	95.203
				R:TACTTGGCAGGGTTTGGTTC		
VvSUT2	VIT_18s0076g00220	XM_002266086.3	Putative sucrose transporter	F:TGACTGGATGGGGAAAGAAG	190	97.956
				R:GTTCCCAATACCCCATACCCG		
VvHT5	VIT_05s0020g03140	NM_001281278.1	Hexose transporter	F:AGCATGAGGAGCTGGAGAGC	198	95.96
				R:CTTGGGCAGCGGTATTAAGC		
VvGBSS1	VIT_02s0025g02790	XM_010661955.2	Granule-bound starch synthase 1, chloroplastic/amyloplastic	F:CCTGGTTCCTTGAGAAGGTATGG	132	93.008
				R:GAATCCGTGGTGCCTCCAGAA		
VvSUS	VIT_11s0016g00470	XM_002275119.3	Sucrose synthase	F:AACTCACCTCTTCTCTAGGTTGTC	200	92.651
				R:GCTAGAAGCTGATGGGGCTG		
VvPSBP1	VIT_12s0028g01080	XM_002283012.4	Probable oxygen-evolving enhancer protein 2	F:CCAACAGCAATGTCTCCGTC	147	93.812
				R:CGTCGAAACCACCCTCATTG		
VvPR10.7	VIT_05s0077g01670	XM_002273754.4	Pathogenesis-related protein 10.7	F:ACCCGGTGTGGAGATCAAAG	180	92.665
				R:AGGCAGCAAGCAACAAGTGA		
VvPR10.3	VIT_05s0077g01550	NM_001281027.1	Pathogenesis-related protein 10.3	F:TGGAGATGTTTTGACGAGCGG	162	94.264
				R:AGAGACTCCTCTTTGCCGCC		
VvETFB	VIT_05s0049g00470	XM_002284716.3	Electron transfer flavoprotein subunit beta, mitochondrial	F:GTCTGGCGTCGGAGGTTATC	165	84.097
				R:CCACATCGACGAGAGCTTTG		
VvTIM13	VIT_02s0025g02990	XM_002277879.3	Mitochondrial import inner membrane translocase subunit Tim13	F:CAAGACTCAGCTTGCCCAGG	201	92.015
				R:GCTCAGTTTCAGCGTGGTGC		
VvNDUFA1	VIT_11s0103g00270	XM_002269070.4	NADH dehydrogenase (ubiquinone) 1 alpha subcomplex subunit 1	F:GCCCGAAGCACGTAGGTAAC	144	91.227
				R:CTTCACACCCAGAAGCACCAAC		

## Data Availability

All data supporting the findings and conclusions of this work are available in the manuscript. Supporting information will be made available from the corresponding author upon request.

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
