# Peer review of "Transcriptomic Analyses of Grapevine Leafroll-Associated Virus 3 Infection in Leaves and Berries of ‘Cabernet Franc’"

_viruses, 2022, doi:10.3390/v14081831_

Round 1

Reviewer 1 Report

The manuscript presented by Song et al. represents a preliminary and interesting piece of work toward the elucidation of the fine molecular interaction between grapevine and GLRaV3.

The applied techniques are appropriate and a widely discussed array of results derived from the experiments.

The paper can be suitable for publication with some minor modifications.

I argue if the title of the paper could be better fitting to the symptomatic target , i.e. the predominant feature of leafroll symptoms in a red berry grape. Even if other viral pathogens affects either GLRD + and - sources, the main issue is focused on sugar metabolism and reddening and curling of leaves. 

The questionable inclusion of GLRaV2 and GLRaV7 in the list of causal agents of leafroll disease could be taken in account, due to several evidences of their null or reduced influence in the etiology of this disease.

Some other points to consider:

94, authors can better discuss elsewhere  that downregulation of anthocyanin biosynthesis (only in berry and not in leaves) is in agreement for what found  in ref. 26 

111, the reference Prator et al (2020) seems to be missed in ref.list.

182, mortar

199-200, a little more information could be added on why and how the FDR has been selected as the main discriminant value for DE

207, The Virtool check for viruses and viroids ( that do not have a polyA tail)  in the RNA-seq dataset could be better discussed, since the cDNA synthesis is supposed to be done through an oligo-dT primer 

221, please explain if a DNAse I digestion has been performed for the gene expression test (or included in the commercial extraction kit) . Even if the primers have been designed across exon-intron joint, could it be excluded any amplification on residual genomic DNA in the total RNA extracts?

241- 245 and  248-249, these sentences repeat something already said in  M&M and do not represent results

246-247, I understand and substantially agree on the definition of 'background infection' about open field-grown grapevine plants. Moreover, the differential key symptom (LR) strongly leads the metabolic pathways reducing a lot the 'noise' effect of minor viruses (if any). But what may happen in disturbing the 'normal' baseline metabolism of a really 'healthy' grape since the presence of two viruses ? Authors may comment this issue briefly in Discussion even if no comparative data can be digged out 

285-288, did the samples LR+ extracted and processed in 2019 (leaves EL38) produce adequate quality and quantity of reads in the RNA-seq similar to the other set done later?

522, DEGs

543, virus titer increase by time ...? at least a reference is needed here. The virus colonization of young vegetation from the old trunk happens any new year  and titer mainly depends by the climatic conditions. 

734-735, chlorosis in white berry grapes and reddening due to anthocyanins in red berry: could the authors step a little on why there is this difference ? 

602,  any evidence for increased expression of the acidic cell wall invertase? 

606, VvHT5

Table S6: remove asterisks at the heading since they are not related with the main captions content below ( I find it misleading)

Author Response

Dear reviewer: 

Please kindly find attached our response to the review report as well as detailed changes made to the manuscript. 

Best 

Yashu

Reviewer 2 Report

This manuscript  described mechanisms that underpin grapevine-GLRaV-3 interactions leading to GLRD through RNA-Seq and global transcriptome analysis.  The results suggest that GLRaV-3 infection may disrupt mitochondrial function in grapevine leaves, leading to repressed sugar export and accumulation of sugar in photosynthetic tissues. The excessive sugar accumulation in GLRaV-3-infected leaves would trigger downstream GLRD symptom development and negatively impact berry quality.  The proposed model provided a clue for the future study of grapevine-GLRaV-3 interaction. Therefore, I recommed this manuscript could be accepted. However, two problems should be clarified.

1. in addition to GLRaV - 3, the other viruses and virods exist, were the effects on grapevine analyzed by RNA-seq caused by GLRaV-3 and other viruses, not only by GLRaV-3?because there were antagonistic or cogenetic relationships between viruses.

2.  why the virus detection results through RNA-seq analysis showed different in different E-L? for example, in Vine No. 15-2 E-L 31 leaf, only GRSPaV and HSVd existed, but in other samples, GRSPaV, GPGV, HSVd and GYSVd1 existed. Does this affect the analysis of the results?

Author Response

(The authors gave the same response as above.)
